# CRISPR prime editing for unconstrained correction of oncogenic *KRAS* variants

Gayoung Jang [1,2], Jiyeon Kweon [1] & Yongsub Kim [1,2✉]

*KRAS* is the most commonly mutated RAS family gene and is a primary cause of the occurrence of several types of cancer. However, *KRAS* mutations have several unique and diverse molecular identities, making it difficult to find specific treatments. Here, we developed universal pegRNAs which can correct all types of G12 and G13 oncogenic *KRAS* mutations with CRISPR-mediated prime editors (PEs). The universal pegRNA successfully corrected 12 types of *KRAS* mutations, accounting for 94% of all known *KRAS* mutations, by up to 54.8% correction frequency in HEK293T/17 cells. We also applied the universal pegRNA to correct endogenous *KRAS* mutations in human cancer cells and found that G13D *KRAS* mutation was successfully corrected to wild-type *KRAS* sequences with up to 40.6% correction frequency without indel mutations. We propose prime editing with the universal pegRNA as a 'one–to–many' potential therapeutic strategy for *KRAS* oncogene variants.

[1] Department of Cell and Genetic Engineering, Asan Medical Institute of Convergence Science and Technology, Asan Medical Center, University of Ulsan College of Medicine, Seoul, Republic of Korea. [2] Stem Cell Immunomodulation Research Center, University of Ulsan College of Medicine, Seoul, Republic of Korea. ✉email: yongsub1.kim@gmail.com

RAS family genes are involved in tumor formation in many human cancers, and there are three *RAS* family genes: *HRAS*, *NRAS*, and *KRAS*[1,2]. Ras proteins are monomeric GTPases and regulate cell differentiation, proliferation, and survival, and can deliver cell growth signals by regulating either guanosine diphosphate (GDP) inactivation or guanosine triphosphate (GTP) activation[3]. When a genetic mutation occurs in RAS, GTP remains active and cell growth signals are sent into the nucleus to induce tumorigenesis. *KRAS* mutations account for most of the *RAS* family gene mutations[4], and intensive efforts have been made over the past 40 years to inhibit the KRAS mutants. Although direct inhibition of KRAS mutants is a desirable approach for treating KRAS-associated tumors, it is still challenging to target mutant KRAS proteins. The structure of KRAS has proven difficult to target due to its smooth surface that hinders the binding by small molecules[5], and a picomolar affinity for GTP, which prevents the development of effective inhibitors[6]. To date, there are two types of inhibitors that have been developed for KRAS in a mutant-specific manner: covalent inhibitors targeting the mutated cysteine residue in KRAS G12C[7–10] and a selective non-covalent inhibitor of KRAS G12D[11,12]. However, direct inhibitors of other KRAS mutations, present in over 50% of KRAS-associated tumors, have not been developed. Therefore, an alternative approach is needed to directly inhibit KRAS mutants.

CRISPR-Cas9 mediated genome editing tools are widely used in a variety of biomedical applications including *KRAS* gene therapy[13–16]. In this system, Cas9 and gRNA complexes of the CRISPR-Cas9 system generate double-strand breaks (DSBs) at target sequences of *KRAS* mutants, and the disrupted *KRAS* mutants inhibit tumor growth[14,15]. However, Cas9 proteins and gRNAs should be carefully adopted for each *KRAS* mutant to reduce unwanted gene disruption in wild-type *KRAS*, and specific gRNAs targeting each *KRAS* mutant should be used for each target mutation (i.e., G12V targeting gRNAs could be used for disrupting only G12V mutation). It has been shown that prolonged Cas9 expression induces other oncogenic *KRAS* variants that are resistant to Cas9 cleavage[16]. CRISPR-Cas9-based base editors (BEs) have been developed to induce base conversions without generating DSBs and have recently been applied to correct *KRAS* G12S, G12D, and G13D mutations[16]. However, each gRNA for base editing should be designed to correct each target

*KRAS* mutation and BEs can correct transition mutations such as C:G to T:A and A:T to G:C as well as transversion mutations such as C:G to A:T or G:C and A:T to C:G or T:A, however, BEs can only correct cytosine or adenine within the base editing window, and often induce undesired bystander substitutions within the base editing window[17–20].

Recently, a revolutionary genome-editing technique, called a prime editor (PE), was developed to induce accurate point mutations in the genome without requiring DSBs or donor DNA templates[21]. PE2 is a fusion protein composed of an engineered reverse transcriptase (RT) and a catalytically inactivated SpCas9 nickase (SpCas9-H840A). PEs are directed to target sites by a prime editing guide RNA (pegRNA) containing primer binding sites (PBS) and a reverse transcriptase template (RTT), which allows the RT to transcribe the additional sequences, which is then introduced at the target sites. The PE3 and PE3b systems increase prime editing efficiency by introducing additional gRNA to nick the non-edited DNA strand, facilitating the edited strand to be used as a repair template.

In this study, we applied the PE system to correct G12 and G13 *KRAS* mutations, which account for most *KRAS* mutations. We designed universal pegRNAs which can correct all types of G12 and G13 *KRAS* mutations and confirmed that the universal pegRNA can correct endogenous *KRAS* mutations in HEK293T/17 cells and three human cancer cell lines. Taken together, these results show the potential of applying the PE system to *KRAS* gene therapy.

## Results

**Design of universal pegRNAs for *KRAS* gene correction.** We first investigated the mutation frequency of three RAS family genes, *HRAS*, *NRAS*, and *KRAS*, which are involved in tumorigenesis. According to the COSMIC (Catalog Of Somatic Mutations In Cancer) database (https://cancer.sanger.ac.uk/)[22], *KRAS* mutations account for the majority (81.4%) of RAS mutations, followed by *NRAS* mutations and *HRAS* mutations, 14.3% and 4.3%, respectively (Fig. 1a). The majority of *KRAS* mutations consist of missense mutations in the G12 and G13 amino acid residues, 80.4% and 13.8%, respectively (Fig. 1a and Table 1).

Because the PE system can introduce various point mutations at target sites using the RTT sequences of pegRNAs, we designed

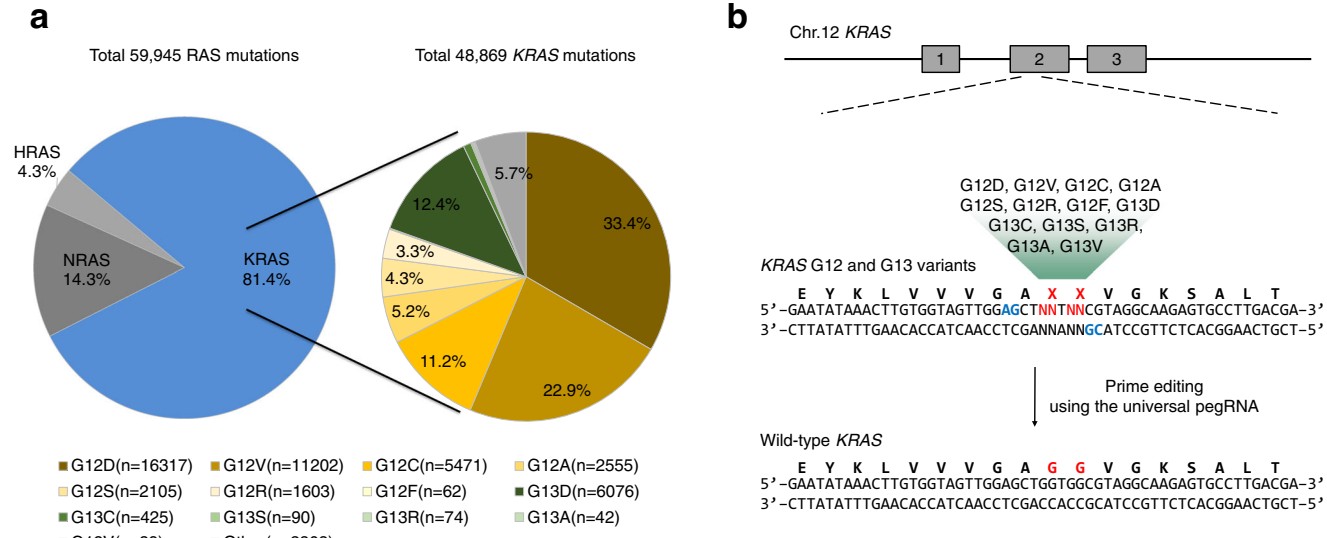

**Fig. 1 Analysis of Ras mutations and design of universal pegRNAs to correct the KRAS mutations. a** Classification of all *RAS* gene family mutations, the left side, and subdivision of *KRAS* mutations, the right side. **b** Schematic overviews of *KRAS* corrections by universal pegRNAs. The G12 and G13 positions of *KRAS* and the PAM sequences of two universal pegRNAs are highlighted in red and blue, respectively.

**Table 1 Targetability of 12 *KRAS* mutations by ABE and PE systems.**

| KRAS mutations | | Counts | Ratio (%) | Targetability | |
| --- | --- | --- | --- | --- | --- |
| Amino acids | Nucleotides | | | ABE | PE |
| p.G12D | c.35 G > A | 16317 | 35.4 | ✓ | ✓ |
| p.G12V | c.35 G > T | 11202 | 24.3 | | ✓ |
| p.G13D | c.38 G > A | 6076 | 13.2 | ✓ | ✓ |
| p.G12C | c.34 G > T | 5471 | 11.9 | | ✓ |
| p.G12A | c.35 G > C | 2555 | 5.5 | | ✓ |
| p.G12S | c.34 G > A | 2105 | 4.6 | ✓ | ✓ |
| p.G12R | c.34 G > C | 1603 | 3.5 | | ✓ |
| p.G13C | c.37 G > T | 425 | 0.9 | | ✓ |
| p.G13S | c.37 G > A | 90 | 0.2 | ✓ | ✓ |
| p.G13R | c.37 G > C | 74 | 0.2 | | ✓ |
| p.G12F | c.34_35delinsTT | 62 | 0.1 | | ✓ |
| p.G13A | c.38 G > C | 42 | 0.1 | | ✓ |
| p.G13V | c.38 G > T | 39 | 0.1 | | ✓ |
| Total counts | | 46,061 | | 24,588 | 46,061 |

universal pegRNAs to manipulate the majority of *KRAS* missense mutations. Previous studies show that prime editing frequencies are dependent on the length between the Cas9-mediated nicking site and the altering site of desired mutation[21,23]. Therefore, we selected two adjacent protospacer sequences with altered NG PAM sequences, *KRAS*-#1 and *KRAS*-#2, to edit G12 and G13 missense mutations and used PAM flexible PE variant, PE2-SpG[24] (Fig. 1b). The length of PBS and RTT in pegRNAs was changed from 9 to 13 bp and 10 to 16 bp, respectively, to generate a 3′ flap of different lengths. In total, we designed nine universal pegRNAs containing RTT sequences to introduce wild-type *KRAS* sequences to the G12 and G13 amino acid residue positions.

**Optimization of universal pegRNAs for *KRAS* gene correction in pooled library.** To evaluate the *KRAS* correction activity of universal pegRNAs, we established HEK293T/17 library cell lines containing 12 selected *KRAS* mutations (HEK293T/17-*KRAS* library). We cloned each of the 12 *KRAS* mutant sequences and barcode sequences into lentiviral plasmids. Lentivirus particles were produced and infected into HEK293T/17 cells with a low multiplicity of infection to introduce only one type of *KRAS* mutation per cell (Fig. 2a). To measure *KRAS* correction activity for each mutation, PCR amplicons were obtained from genomic DNA and subjected to targeted deep sequencing analysis. Using the barcode sequence of the amplicon, sequencing results were classified as the original *KRAS* mutations and the *KRAS* correction activity of each mutation was calculated. Using the HEK293T/17-*KRAS* library cell lines, we examined *KRAS* correction activities of universal pegRNAs using PE2-SpG and revealed that universal pegRNAs including *KRAS*-#2 protospacer sequences could correct all 12 types of *KRAS* mutations with higher activity compared to universal pegRNAs including *KRAS*-#1 protospacer sequences (Fig. 2b and Supplementary Fig. 1). The pegRNAs with longer RTTs tended to have higher *KRAS* correction activities and a pegRNA including 13-nt PBS and 16-nt RTT had 28.3% *KRAS* G12S correction activity. We also found that none of the pegRNAs with PE2-SpG induced any substitutions or indels at endogenous *KRAS* sites (Supplementary Table 1).

To further improve the *KRAS* correction efficiency, the universal pegRNA containing 13-nt PBS and 16-nt RTT was subjected to the PE3b system (hereafter named, PE3b-SpG) which uses additional gRNAs to nick the opposite strand of the edited strand to increase the editing frequency[21]. As shown in Fig. 2c,

the *KRAS* correction activity was improved up to 2.6-fold in the G12R mutation by the PE3b-SpG, and in the G12R *KRAS* mutation, the correction efficiency was improved to 50.1% without any mutations at endogenous *KRAS* sites. In addition, we further constructed engineered pegRNAs (epegRNAs) containing either tevopreQ1 or tmpknot RNA motif at the 3′ end of pegRNAs, which is known to enhance prime editing activity[25]. Compared with the original universal pegRNA, universal epegRNAs had improved *KRAS* correction activities, which were further enhanced by PE3b-SpG. We found that the universal epegRNA containing tevopreQ1 RNA motif could edit *KRAS* G12C and G13C mutation by 54.8% (Fig. 2c). We also confirmed that there were no substitutions or indel mutations at endogenous *KRAS* sites (Supplementary Table 1). Overall, these results indicated that the universal pegRNA could correct the majority of *KRAS* mutations, and epegRNA and PE3b-SpG could further improve the *KRAS* correction efficiency without altering the wild-type *KRAS* sequence.

**Correction of endogenous *KRAS* mutations using universal pegRNAs.** We then decided to correct endogenous *KRAS* mutations using the universal epegRNA. To generate cell lines containing endogenous *KRAS* mutations, we designed several pegRNAs to introduce *KRAS* G12D, G12V, G13C, or G13D mutations at the wild-type endogenous *KRAS* sequence (Fig. 3a). We delivered plasmids encoding the pegRNAs and PE2 into HEK293T/17 cells and prime editing frequencies were measured by targeted-deep sequencing. As shown in Supplementary Fig. 2, prime editing frequencies varied from 0.3% to 36.0% and we selected pegRNAs with the highest prime editing frequency per target *KRAS* mutation. We then transfected the plasmid encoding each pegRNA and PE2 into HEK293T/17 cells and isolated single clones containing *KRAS* G12D, G12V, G13C, or G13D mutations at the endogenous *KRAS* gene. The Sanger sequencing results showed that each clone had heterozygous *KRAS* mutations (Fig. 3b). As the *KRAS* gene is located on chromosome 12, which exists as a triploid in HEK293T/17 cells[26], each clone had heterogenous *KRAS* sequences, with two copies of the wild-type *KRAS* sequence and one copy of the mutated *KRAS* sequence.

To measure the endogenous *KRAS* correction frequency, we delivered the universal pegRNA containing 13-nt PBS and 16-nt RTT and PE2-SpG into each HEK293T/17-*KRAS* G12D, G12V, G13C, and G13D cell line. We also examined epegRNAs and PE3b-SpG to determine whether endogenous *KRAS* correction frequency could be enhanced. The *KRAS* correction frequency of *KRAS* heterogenous cells was calculated using the reads counts from targeted-deep sequencing as described in the methods section. As in HEK293T/17-*KRAS* library cell lines in Fig. 2b, c, we identified that the PE3b-SpG showed higher *KRAS* correction activity than the PE2-SpG in four endogenous *KRAS* mutated HEK293T/17 cells (Fig. 3c and Supplementary Fig. 3). The tevopreQ1 and tmpknot RNA motifs could improve *KRAS* correction activity, and the epegRNA containing tmpknot RNA motif could correct *KRAS* G13D mutation by 31.6% in PE3b-SpG. We found that there were no detectable indel mutations in the heterogenous cell lines (Supplementary Table 2). We also constructed a universal pegRNA capable of inducing a silent mutation to confirm that the *KRAS* corrections were indeed induced by prime editing and confirmed that the *KRAS* corrected alleles also contained the silent mutation (Supplementary Fig. 4).

Furthermore, we examined whether the universal pegRNA could correct the endogenous *KRAS* mutations in three types of human cancer cells: two pancreatic cancer cell lines, CFPAC-1 and ASPC-1, and one colon cancer cell line, HCT116[27,28]. We confirmed that CFPAC-1 cells have three *KRAS* G12V alleles

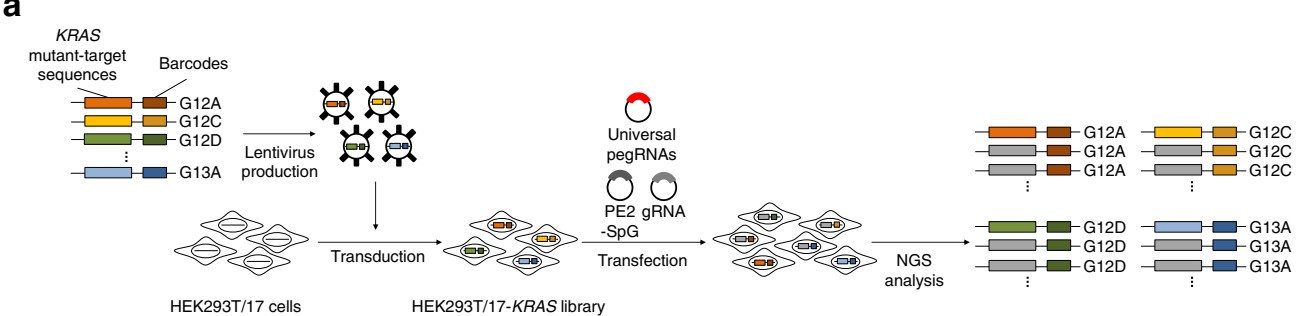

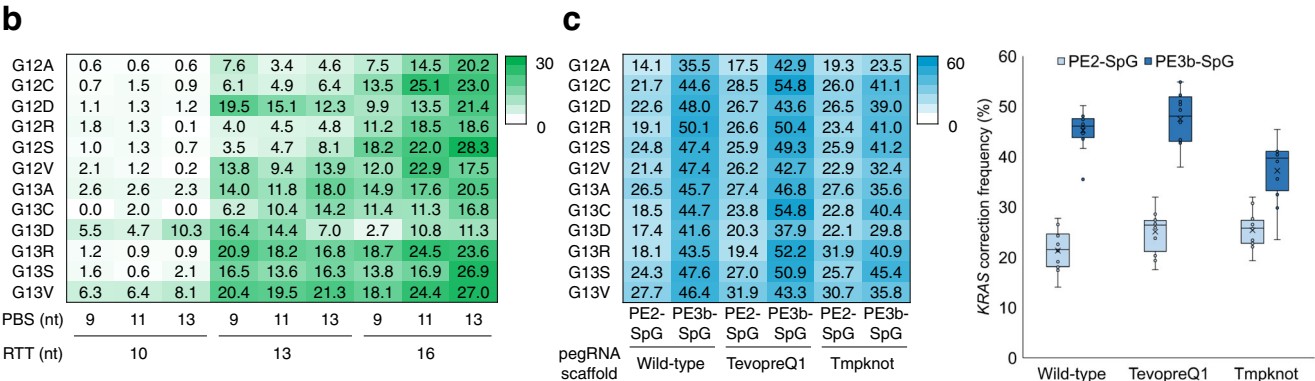

**Fig. 2 Construction of HEK293T/17-KRAS library cells and correction of KRAS mutations using universal pegRNAs. a** Schematic overview for correcting *KRAS* mutations in HEK293T/17-*KRAS* library cells. Lentivirus particles containing 12 types of *KRAS* mutant-target sequences and barcode sequences were delivered in HEK293T/17 cells to generate HEK293T/17-*KRAS* library cells. After editing *KRAS* mutations using universal pegRNAs, the *KRAS*-corrected cells were collected and subjected to NGS analysis to measure the *KRAS* correction frequencies of each *KRAS* mutation. **b** Nine universal pegRNAs containing various lengths of PBS and RTT and PE2-SpG were delivered in HEK293T/17-*KRAS* library cells, and the *KRAS* correction frequency of each *KRAS* mutation is shown in the heatmap. **c** Universal pegRNAs containing 13-nt PBS and 16-nt RTT were engineered to have two types of RNA motif, tevopreQ1 and tmpknot, and subjected to *KRAS* correction in PE2-SpG or PE3b-SpG systems. The *KRAS* correction frequencies are shown in the heatmap and box-whisker plot, respectively. The experiments were conducted with biological triplicate ($n = 3$).

and one *KRAS* wild-type allele, ASPC-1 cells have two *KRAS* G12D mutation-bearing alleles, and HCT116 cells have one *KRAS* G13D allele and one *KRAS* wild-type allele by targeted-deep sequencing. We first examined the *KRAS* correction activity of the universal epegRNA with tevopreQ1 motif in PE3b-SpG, but the *KRAS* correction activity was only 2.7% in ASPC-1 cells, much lower than that of the HEK293T/17 cells (Fig. 3d). To further improve the prime editing frequency, we applied PE5max system[29], which utilized PEmax-SpG-P2A-hmLH1dn (hereafter named, PE5max-SpG) instead of PE2-SpG, and found that the *KRAS* mutations in 13.2% of CFPAC-1 cells and 18.7% of ASPC-1 cells were corrected by the universal epegRNAs (Fig. 3d, e). In the case of HCT116, PE3b-SpG showed 32.0% *KRAS* correction frequency and PEmax-SpG (hereafter named, PE4max-SpG) showed 36.1% *KRAS* correction frequency (Fig. 3d, e). We measured the KRAS correction frequency for two weeks to evaluate the functional effect of *KRAS* gene correction and found that the *KRAS* correction frequency decreased over time, reflecting that the *KRAS* corrected HCT116 cells have a growth disadvantage (Fig. 3f). Taken together, we demonstrated that endogenous *KRAS* mutations could be corrected by the universal pegRNA in HEK293T/17 cells, pancreatic cancer cell lines, and colon cancer cell lines.

## Discussion

Although *KRAS* mutations are the most frequent oncogenic alterations, these mutations have remained an intractable

therapeutic target. In this study, we developed the universal pegRNA to correct most types of *KRAS* mutations and corrected these *KRAS* mutations without unwanted mutations using the PE system in the library cells. Furthermore, we successfully corrected endogenous *KRAS* mutations in human cells.

Previously, the Cas9 nuclease and base editors were successfully applied to target *KRAS*[14,15]. Cas9 nuclease with mutant–specific gRNAs enable *KRAS* mutant-specific disruption. Although Cas9 nuclease disrupts *KRAS* mutations with high efficiency, this method requires careful selection of mutant–specific gRNAs and Cas9 protein at each *KRAS* variant[16]. The adenine base editors which introduce A:T to G:C conversion in the target locus could also correct several types of *KRAS* mutations including G12S, G12D, and G13D, however, gRNAs for each mutation should be designed individually for the BE-mediated correction of *KRAS* mutations. Furthermore, adenine base editors can target only 4 types of *KRAS* mutations in *KRAS* G12 and G13 mutations (Table 1). As unwanted bystander editing often occurs in BE system due to the base editing active window[30], it is necessary to confirm the performance of each gRNA, which is laborious.

The PE system has advantages over the Cas9 nuclease and base editors in editing *KRAS* mutations. Theoretically, the PE system can manipulate all types of *KRAS* G12 and G13 mutations using only one type of pegRNA, making it simple to use in biomedical applications. Although the PE system has advantages in correcting *KRAS* mutations, the correction efficiencies of two pancreatic

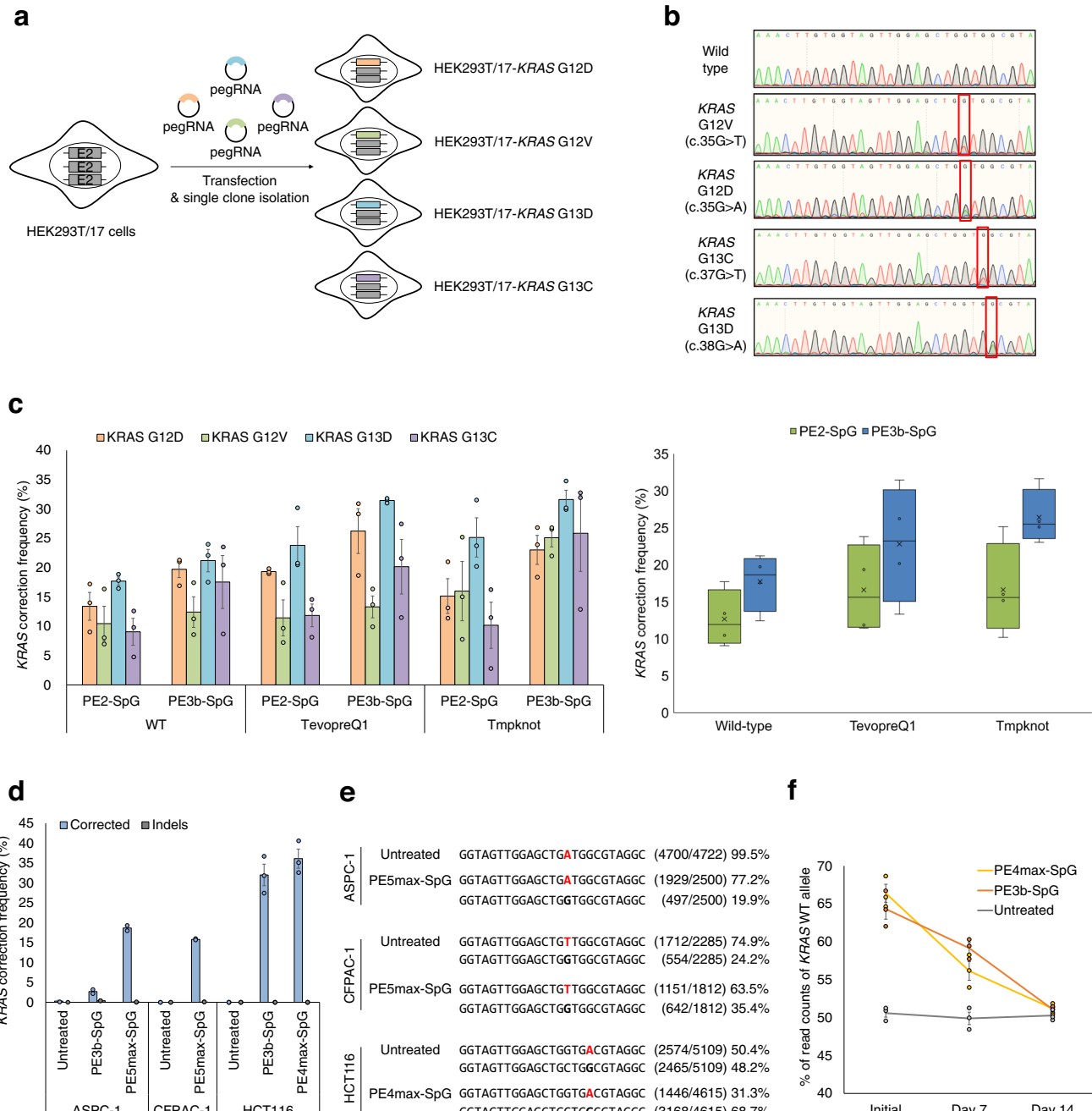

**Fig. 3 Correction of endogenous KRAS mutations in HEK293T/17 cells and three human cancer cell lines. a** Schematic for the construction of HEK293T/17 cell lines containing four types of *KRAS* mutations at endogenous *KRAS* sites. The pegRNAs and PE2 were delivered into HEK293T/17 cells and single clones were analyzed to obtain HEK293T/17 cells with endogenous *KRAS* mutations. **b** Sanger sequencing results for each HEK293T/17 cell line containing endogenous *KRAS* mutations. **c** Endogenous *KRAS* correction frequency in *KRAS* mutated-HEK293T/17 cells. The universal pegRNAs and epegRNAs were delivered into each *KRAS* mutated-HEK293T/17 cell by PE2-SpG or PE3b-SpG systems. The *KRAS* correction frequency was calculated from read counts of NGS sequencing results described in Table S2. Error bars mean s.e.m. of biological triplicate samples (*n* = 3). **d** Endogenous *KRAS* correction in three human cancer cell lines, ASPC-1, CFPAC-1 and HCT116 cells. The *KRAS* correction frequencies of *KRAS* heterogenous cell lines, CFPAC-1 and HCT116, were calculated from read counts of NGS sequencing results. Error bars mean s.e.m. of biological duplicate or triplicate samples. **e** The representative NGS sequencing results of *KRAS* correction in three human cancer cell lines. *KRAS* mutations in each cell line (G12D, 12 V, and G13D) were highlighted in red. **f** The *KRAS* correction frequency of HCT116 was measured over time. Error bars mean s.e.m. of biological triplicate samples (*n* = 3).

cancer cell lines were much lower than those of HEK293T/17 cell lines and HCT116 colon cancer cells as previously described[29]. As low editing frequencies made it difficult to detect the effects of *KRAS* gene correction, such as aberrant growth rate and functional analysis of *KRAS* downstream pathways, It is necessary to improve the prime editing activity in vitro and in vivo for the PE system to be used in therapeutic approaches. Previously, it has shown that the PE system does not induce appreciable off-target mutations, however, off-target effects should be considered to be used in gene therapy strategy[31–34]. We selected five potential off-target sites based on the number of mismatches with the universal epegRNA and analyzed whether mutation was induced at those

five sites (Supplementary Table 3). We confirmed that there was no remarkable mutation at potential off-target sites by targeted-deep sequencing, and high-throughput analysis, such as transcriptome analysis, might be necessary to elucidate the potential off-target effects (Supplementary Fig. 5). Our finding showed that the PE system could introduce and correct the *KRAS* mutations, thereby demonstrating the potential of the system for developing *KRAS*-specific targeted therapies.

## Methods

**Cell culture and cell line generation**. CFPAC-1 cells and ASPC-1 cells were gifted by Dr. Eunsung Jun (Asan Medical Center) and maintained in Roswell Park Memorial Institute 1640 Medium and HEK293T/17 cells (ATCC CRL-11268) and HCT116 cells (ATCC CCL-247) were maintained in Dulbecco's modified Eagle's medium, supplemented with 10% fetal bovine serum and 1% penicillin/streptomycin.

To generate the HEK293T-*KRAS* library stable cell line, lentiviral particles were generated and transduced into HEK293T/17 cells. Briefly, 1 μg of plasmids (500 ng of *KRAS* mutant sequence containing vectors, 300 ng of psPAX2, and 200 ng of pMD2.G) were transfected into $2 \times 10^5$ HEK293T/17 cells using Lipofectamine 2000 (Thermo Fisher Scientific) by the manufacturer's instructions, and the medium was changed 24 h after transfection[23]. The supernatants containing lentiviral particles were harvested and filtered with 0.45 μm filter 48 h after transfection and stored at –70 °C until use. The lentivirus particles were then transduced into HEK293T/17 cells with a low multiplicity of infection and infected cells were selected for 3 days with 1 μg/ml of puromycin. The *KRAS* mutant sequences were listed in Supplementary Table 4.

**Plasmid construction**. The pRG2 plasmid (Addgene plasmids #104174) was used for gRNA cloning and pU6-pegRNAGG-acceptor (Addgene plasmids #132777), pU6-tevopreq1-GG-acceptor (Addgene plasmids #174038), and pU6-tmpknot-GG-acceptor (Addgene plasmids #174039) plasmids were used for pegRNA or epegRNA cloning. The PCR amplicon of epegRNAs and PE3b gRNAs were cloned into the lentiGuide-puro plasmid (Addgene plasmids #52963) to construct an all-in-one vector for use in human cancer cells. To construct a lentiviral vector containing *KRAS* mutant target sequences to generate HEK293T/17-*KRAS* library stable cell lines, *KRAS* mutant target sequences were cloned into pCW-Cas9 plasmids (Addgene # 50661). The sequences of target sequences, PBS, and RTT are listed in Supplementary Table 5. The pCMV-PE2-SpG-V3, which was constructed by fusing two nuclear localization signals to both of C- and N-terminal of pCMV-PE2-SpG (Addgene plasmid #159978), and pCMV-PEmax-SpG-P2A-hMLH1dn, which was constructed by site-directed mutagenesis from pCMV-PEmax-P2A-hMLH1dn (Addgene plasmid #174828), were used for prime editing experiments.

**Transfection**. For the plasmid transfections, HEK293T/17 cells ($0.6 \times 10^5$), HEK293T/17-*KRAS* library cells ($0.6 \times 10^5$), CFPAC-1 cells ($3 \times 10^4$), ASPC-1 cells ($3 \times 10^4$), or HCT116 cells ($2 \times 10^4$) were seeded onto a TC-treated 48-well plate and transfection was performed the next day when cell confluency reached 50–60%. A total of 500 ng of plasmids (250 ng of PEs with 250 ng of pegRNAs or epegRNAs) were delivered using 1.5 μL of Lipofectamine 2000 (Thermo Fisher Scientific) or 1.5 μL Lipofectamine 3000 (Thermo Fisher Scientific) with 1 μL of P3000 reagent according to the manufacturer's protocol. For the PE3b experiments in HEK293T/17 cells, 83 ng of plasmids encoding gRNAs were additionally delivered to the cells. For the PE3b experiments in CFPAC-1, ASPC-1, or HCT116 cells, 250 ng of all-in-one vector was used for the plasmid transfection, and transfected cells were selected one day after transfection with 1 μg/mL of puromycin.

**Targeted-deep sequencing**. Genomic DNA was extracted 72 h after transfection using cell lysis buffer (0.05% SDS in pH 7.5 of 100 nM Tris-HCl and 100 μg/ml proteinase K) and subjected to PCR amplification. The target sites were amplified using Phusion High-Fidelity DNA Polymerase (New England Biolabs) with target-specific primer pairs and the Illumina TruSeq HT dual index adaptor primer (Supplementary Table 6)[35]. The libraries were sequenced at 150-bp paired-end using the Illumina MiniSeq or iSeq 100 sequencing equipment to analyze the mutation frequencies and Cas-Analyzer (http://www.rgenome.net/cas-analyzer)[36] was used to analyze the sequencing data. *KRAS* correction frequencies in homozygous *KRAS* mutant cells including HEK293T/17-*KRAS* library cell lines and ASPC-1 cells were directly measured as prime editing frequencies, and those in heterozygous *KRAS* mutant cells including HEK293T/17 mutant clones, CFPAC-1 and HCT116 cells, were calculated as follows:

$$KRAS\ correction\ frequency(\%) = \frac{\%\ of\ KRAS\ WT\ in\ treated\ cells - \%\ of\ KRAS\ WT\ in\ untreated\ cells}{\%\ of\ KRAS\ mutants\ in\ untreated\ cells}$$

(1)

**Sanger sequencing**. Genomic DNA was isolated using the DNA Blood & Tissue kit (Qiagen) in accordance with the manufacturer's instructions and target sites were amplified using phusion high-fidelity DNA polymerase (New England Biolabs) to verify the endogenous *KRAS* sequences of single clones. The PCR products were purified with the QIAquick PCR Purification Kit (Qiagen) and analyzed by Sanger sequencing.

**Statistics and reproducibility**. All statistical analyzes were performed independently on two or three biological experiments. Error bars represent standard error of the mean.

**Reporting summary**. Further information on research design is available in the Nature Portfolio Reporting Summary linked to this article.

## Data availability

DNA sequencing data have been deposited in the National Center for Biotechnology Information (NCBI) Sequence Read Archive (SRA) database with BioProject accession code PRJNA972834. Source data for the plots and graphs in the figures is available as Supplementary Data 1 and any remaining information can be obtained from the corresponding author upon reasonable request.

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

## Acknowledgements

We thank Dr. Eunsung Jun (Asan Medical Center) for providing CFPAC-1 cells and ASPC-1 cells. Y.K. supervised the research. G.J. and J.K. performed the experiments and analyzed the data. G.J., J.K. and Y.K. wrote the manuscript. All authors approved the manuscript. This work was supported by the National Research Foundation of Korea [2021R1C1C100716212, 2018R1A5A2020732 to Y.K.].

## Author contributions

G.J. and Y.K. conceived and designed this study. G.J. and J.K. carried out the experiments and analyzed the data. G.J. and Y.K. wrote the manuscript. Y.K. supervised the research.

## Competing interests

The authors declare no competing interests.
