## [Peer Review File · Communications Biology]

Reviewers' comments:

Reviewer #1 (Remarks to the Author):

The manuscript by Jang et al. describes a prime editing (PE) approach to correct G12 and G13 oncogenic KRAS mutations. The experimental approach includes devising a clever assay to install 12 selected KRAS variants through a barcoded-lentiviral library in HEK293T cells, before transfecting PE plasmids and a panel of pegRNAs. NGS analysis revealed editing efficiencies of various KRAS mutations reaching up to 54.7% in HEK293T/17 cells and up to 18.7% in pancreatic cell lines. While there is room for improvement of editing and delivery efficiencies, I find the work novel and it adds to the arsenal of new strategies to target KRAS. In general, the paper is well-written and the data fits its main conclusions, however, there are a few comments/suggestions that should be addressed:

- In Figure 3E editing in CFPAC-1 cells seems to reach correction rates of around 15%, however, sequencing in panel 3F, shows ~11% editing. Is there an explanation for this discrepancy?
- The authors conclude that the low editing efficiency in pancreatic cells precludes the phenotypic consequence of KRAS correction such as growth retardation/inactivation of MAPK. I would be intrigued to see a bit of mechanistic insight and validate KRAS correction by a functional consequence for its inactivation. Therefore, one suggestion I have would be to test the correction in readily transfectable KRAS-mutant cells (e.g., HCT116, PANC-1, A549).
- Towards measuring an effect after KRAS correction, the authors could isolate DNA samples from treated cells for an extended time period. Determining the corrected KRAS frequency from the DNA samples over time might should reveal a growth disadvantage of the corrected cells. I would expect that the non-corrected cells overgrow the corrected cells, which would be reflected in declining correction rates in samples collected at later time points.
- In all experiments, the editing can only be as good as the delivery of prime editing components: I am missing a transfection control, especially in the pancreatic cancer cell lines.

Reviewer #2 (Remarks to the Author):

Jang et al. investigate prime editing, a recently developed genome editing technology, as a strategy for correction of oncogenic KRAS variants in cell-based systems. The authors develop a suite of universal pegRNAs to correct G12 and G13 mutations in KRAS in a pooled mutant 293T library, single endogenous mutant 293T cells, and G12V/D mutant cancer cell lines. Impressive prime editing outcomes are observed, and the study also provides some valuable insights into use of NG-PAM prime editors and pegRNA parameters. The authors rely on published work to optimize pegRNA editing efficiency, including PE3b, PEMax, and epegRNA strategies. As the authors note, correction of oncogenic KRAS mutations using gene editing is not an altogether novel strategy, though the application of prime editing does offer clear advantages, which they highlight. The rationale was clearly stated, and experiments were logically performed. The manuscript does suffer from some less-than-clear language, factual inaccuracies, incomplete citation of the relevant literature, and inconsistent Figure references, which made evaluation of the data more challenging.

Beyond the limitations listed below, the overall impact of this study is fairly limited with respect to prime editor-related technology advancement, cancer genetics, or genome engineering-focused cancer therapy.

Major concerns:

Figure references are incorrect consistently throughout the paper.

It is the accepted standard in genome editing literature to performing sequencing experiments in triplicate. Numbers of at least n=3 should be used to evaluate pegRNA efficiency.

The authors show that 293 cells can be prime edited at the endogenous locus to correct KRAS mutations. However, additional mechanisms independent of prime editing could correct the mutant allele. For example, gRNA nicking may promote WT strand invasion and mutant repair. Introduction of silent edits that revert KRAS mutants to WT would prove that mutant correction is due to prime editing.

While several studies suggest that prime editing does not lead to appreciable off target effects, there is no mention of this throughout the manuscript. As the authors propose use of prime editing as a gene therapy strategy, this must be at least discussed. Off-target editing would ideally be investigated experimentally, but the authors should at least cite relevant literature on this topic.

Given the relatively low editing outcomes in KRAS-driven cancer cell lines, it is difficult to imagine this application of prime editing as a viable therapeutic strategy in patients. It is highly likely that unedited cells would outcompete edited cells over time. Even in cell culture, one suspects that after sustained culture (>72h post transfection), the fraction of corrected cells would be significantly reduced due to positive selection of KRAS mutant cells. This is a major challenge to the implication of the authors' work and its broader impact on cancer biology and therapy. While demonstration of in vivo efficacy may be beyond the scope of this study, the authors should consider performing experiments to evaluate the functional effects of KRAS gene correction. This could include cell viability or growth assays or evaluating cells by microscopy to identify morphologic differences or apoptotic cells in the context of KRAS mutant correction.

Minor concerns:

It is unclear what is meant by "RAS combines with guanosine triphosphatase (GTPase)". RAS proteins are small GTPases. Please clarify what is meant here.

The statement "KRAS has a structure lacking a pocket that can bind to allosteric inhibitors" is incorrect and is directly contradicted by literature that the authors cite in the following sentence (Ostrem et al).

The statement "BEs can only correct transition mutations such as C:G to T:A and A:T to G:C conversion" is incorrect. Several groups have published transversion base editors. While these are not perfect tools, their omission is misleading.

The text suggests that CFPAC-1 cells have 4 KRAS alleles, 1 of which is KRAS G12V mutant. This would seem inconsistent with their untreated sequencing data in Figure 3F? Please reconcile this.

Figure 2 demonstrates consistent increases in prime editing efficiency through use of a PE3b strategy, including a 3.3 fold increase in the correction of the G13D mutation. The authors suggest that no mutations at the endogenous gene were observed, but do not show unintended edits for the target site. Indel frequency and unintended editing must be shown, particularly given the rate of 'unwanted mutations' at the edit site in Supplementary Figure 3.

In Supplementary Figure 1, editing data are missing for 6 KRAS mutations.

Unintended editing outcomes should be shown in Supplementary Figure 2.

It is unclear what is meant by 'unwanted mutation' in Supplementary Figure 3. Does this include indels?

In Supplementary Figure 3, changes in KRAS allele frequency are subtle in some cases. Appropriate statistical tests should be used to demonstrate the effect of prime editing.

Point-by-point response

We would like to thank the reviewers for carefully reading our manuscript and providing valuable comments to improve it. We have now addressed various issues raised by the reviewers as specified below and highlighted textual changes in our revised manuscript for ease of tracking.

Reviewer #1:

The manuscript by Jang et al. describes a prime editing (PE) approach to correct G12 and G13 oncogenic KRAS mutations. The experimental approach includes devising a clever assay to install 12 selected KRAS variants through a barcoded-lentiviral library in HEK293T cells, before transfecting PE plasmids and a panel of pegRNAs. NGS analysis revealed editing efficiencies of various KRAS mutations reaching up to 54.7% in HEK293T/17 cells and up to 18.7% in pancreatic cell lines. While there is room for improvement of editing and delivery efficiencies, I find the work novel and it adds to the arsenal of new strategies to target KRAS. In general, the paper is well-written and the data fits its main conclusions, however, there are a few comments/suggestions that should be addressed:

Response: We appreciate this reviewer's comprehensive summary with supportive comments and have carefully revised all the points raised as follows.

- In Figure 3E editing in CFPAC-1 cells seems to reach correction rates of around 15%, however, sequencing in panel 3F, shows ~11% editing. Is there an explanation for this discrepancy?

Response: We understood the reviewer's confusion regarding to the difference. We differently calculated *KRAS* correction frequencies of *KRAS* homozygous and heterozygous cells as described in Methods section. To provide a better understanding to readers, we have now described them in the RESULTS section as below.

Page 6, line 4:

"The KRAS correction frequency of KRAS heterogenous cells was calculated using the reads counts from targeted-deep sequencing as described in the methods section."

- The authors conclude that the low editing efficiency in pancreatic cells precludes the phenotypic consequence of KRAS correction such as growth retardation/inactivation of MAPK. I would be intrigued to see a bit of mechanistic insight and validate KRAS correction by a functional consequence for its inactivation. Therefore, one suggestion I have would be to test the correction in readily transfectable KRAS-mutant cells (e.g., HCT116, PANC-1, A549).

Response: We appreciate the reviewer's valuable suggestion for improving our manuscript. We have now examined the universal epegRNA in HCT116 cells and described the results in Figure 3D and 3E and in the RESULTS section as highlighted.

Figure 3D and 3E:

• Towards measuring an effect after KRAS correction, the authors could isolate DNA samples from treated cells for an extended time period. Determining the corrected KRAS frequency from the DNA samples over time might should reveal a growth disadvantage of the corrected cells. I would expect that the non-corrected cells overgrow the corrected cells, which would be reflected in declining correction rates in samples collected at later time points.

Response: We appreciate the reviewer's valuable comments for improving our manuscript. We have now measured the *KRAS* correction frequency of HCT116 cells over time and described the result in Figure 3F and in the RESULTS section as below.

Figure 3F:

Page 6, line 27:

“We measured the KRAS correction frequency for two weeks to evaluate the functional effect of KRAS gene correction and found that the KRAS correction frequency decreased over time, reflecting that the KRAS corrected HCT116 cells have a growth disadvantage (Figure 3F)”

- In all experiments, the editing can only be as good as the delivery of prime editing components: I am missing a transfection control, especially in the pancreatic cancer cell lines.

Response: We agree with the reviewer that transfection efficiency affects prime editing frequency. We first confirmed the transfection efficiency of pancreatic cancer cell lines through the expression levels of GFP. As we found that the pancreatic cancer cells had a low transfection efficiency and cell viability compared to HEK293T/17 cells, we used plasmids encoding the puromycin resistance gene and selected transfected cells with puromycin. We have now mentioned the detailed transfection protocol of human cancer cells (including HCT116) in the Methods section as highlighted.

Page 8, line 14:

“The PCR amplicon of epegRNAs and PE3b gRNAs were cloned into the lentiGuide-puro plasmid (Addgene plasmids #52963) to construct an all-in-one vector for use in human cancer cells.”

Page 8, line 33:

“For the PE3b experiments in CFPAC-1, ASPC-1, or HCT116 cells, 250 ng of all-in-one vector was used for the plasmid transfection, and transfected cells were selected one day after transfection with 1 µg/mL of puromycin.”

Reviewer #2:

Jang et al. investigate prime editing, a recently developed genome editing technology, as a strategy for correction of oncogenic KRAS variants in cell-based systems. The authors develop a suite of universal pegRNAs to correct G12 and G13 mutations in KRAS in a pooled mutant 293T library, single endogenous mutant 293T cells, and G12V/D mutant cancer cell lines. Impressive prime editing outcomes are observed, and the study also provides some valuable insights into use of NG-PAM prime editors and pegRNA parameters. The authors rely on published work to optimize pegRNA editing efficiency, including PE3b, PEmax, and epegRNA strategies. As the authors note, correction of oncogenic KRAS mutations using gene editing is not an altogether novel strategy, though the

application of prime editing does offer clear advantages, which they highlight. The rationale was clearly stated, and experiments were logically performed. The manuscript does suffer from some less-than-clear language, factual inaccuracies, incomplete citation of the relevant literature, and inconsistent Figure references, which made evaluation of the data more challenging. Beyond the limitations listed below, the overall impact of this study is fairly limited with respect to prime editor-related technology advancement, cancer genetics, or genome engineering-focused cancer therapy.

Response: We appreciate this reviewer's comprehensive summary with supportive comments and have carefully revised all the points raised as follows.

Major concerns:

Figure references are incorrect consistently throughout the paper.

Response: We apologize for our mistakes and have now corrected the Figure references throughout the manuscript.

It is the accepted standard in genome editing literature to performing sequencing experiments in triplicate. Numbers of at least $n=3$ should be used to evaluate pegRNA efficiency.

Response: We have now evaluated the activity of pegRNAs with biological triplicate ($n=3$) and revised throughout the manuscript.

The authors show that 293 cells can be prime edited at the endogenous locus to correct KRAS mutations. However, additional mechanisms independent of prime editing could correct the mutant allele. For example, gRNA nicking may promote WT strand invasion and mutant repair. Introduction of silent edits that revert KRAS mutants to WT would prove that mutant correction is due to prime editing.

Response: We appreciate the reviewer's valuable comments for improving our manuscript. As the reviewer's suggestion, we additionally constructed a universal pegRNA capable of inducing silent mutation as well as correcting *KRAS* mutation. We transfected them into HEK293T/17-KRAS G12V heterogenous cells and analyzed the sequences of alleles by targeted-deep sequencing. We confirmed that *KRAS* corrected allele was accompanied by the silent mutation and described the results in Supplementary Figure 4.

Supplementary Figure 4:

KRAS G12V mutant alleles of HEK293T/17 heterogenous cells

E Y K L V V V G A V G V G K S A L T
 5'-GAATATAAACTTGTGGTAGTTGGAGCTGTGGCGTAGGCAAGAGTGCCTTGACGA-3'
 3'-CTTATATTTGAACACCATCAACCTCGACAACCGCATCCGTTCTCACGGAACTGCT-5'

↓
 Prime editing
 using the universal pegRNA
 containing a silent mutation

E Y K L V V V G A G G V G K S A L T
 5'-GAATATAAACTTGTGGTAGTTGGAGCAGGTGGCGTAGGCAAGAGTGCCTTGACGA-3'
 3'-CTTATATTTGAACACCATCAACCTCGTCCACCGCATCCGTTCTCACGGAACTGCT-5'

Untreated	TAGTTGGAGCTG G TGGCGTAGGCAA	(2434/4029)	60.4%
	TAGTTGGAGCTG T TGGCGTAGGCAA	(1519/4029)	37.7%
	TAGTTGGAGC A G G TGGCGTAGGCAA	(0/4029)	0.0%
	TAGTTGGAGC A G T TGGCGTAGGCAA	(0/4029)	0.0%
pegRNA treated	TAGTTGGAGCTG G TGGCGTAGGCAA	(2889/4738)	61.0%
	TAGTTGGAGCTG T TGGCGTAGGCAA	(1561/4738)	32.9%
	TAGTTGGAGC A G G TGGCGTAGGCAA	(206/4738)	4.3%
	TAGTTGGAGC A G T TGGCGTAGGCAA	(0/4738)	0.0%

Page 6, line 12:

“We also constructed a universal pegRNA capable of inducing a silent mutation to confirm that the KRAS corrections were indeed induced by prime editing and confirmed that the KRAS corrected alleles also contained the silent mutation (Supplementary Figure 4).”

While several studies suggest that prime editing does not lead to appreciable off target effects, there is no mention of this throughout the manuscript. As the authors propose use of prime editing as a gene therapy strategy, this must be at least discussed. Off-target editing would ideally be investigated experimentally, but the authors should at least cite relevant literature on this topic.

Response: We appreciate the reviewer’s valuable comments for improving our manuscript. We have now examined whether the universal epegRNA had off-target effects. We selected top 5 potential off-target sites based on the number of mismatches with the spacer sequence, and confirmed whether mutations were introduced at those sites by targeted-deep sequencing. We found that no off-target mutations were induced at five potential off-target sites, and described the results in the Supplementary Figure 5 and Supplementary Table 3. We have discussed the off-target effects of PE system in the DISCUSSION section as below.

Page 7, line 20:

“Previously, it has shown that the PE system does not induce appreciable off-target mutations, however, off-target effects should be considered to be used in gene therapy strategy . We selected five potential off-target sites based on the number of mismatches with the universal epegRNA and analyzed whether mutation was induced at those five sites (Supplementary Table 3). We confirmed that there was no

remarkable mutation at potential off-target sites by targeted-deep sequencing, and high-throughput analysis, such as transcriptome analysis, might be necessary to elucidate the potential off-target effects (Supplementary Figure 5).”

Supplementary Figure 5:

Given the relatively low editing outcomes in KRAS-driven cancer cell lines, it is difficult to imagine this application of prime editing as a viable therapeutic strategy in patients. It is highly likely that unedited cells would outcompete edited cells over time. Even in cell culture, one suspects that after sustained culture (>72h post transfection), the fraction of corrected cells would be significantly reduced due to positive selection of KRAS mutant cells. This is a major challenge to the implication of the authors' work and its broader impact on cancer biology and therapy. While demonstration of in vivo efficacy may be beyond the scope of this study, the authors should consider performing experiments to evaluate the functional effects of KRAS gene correction. This could include cell viability or growth assays or evaluating cells by microscopy to identify morphologic differences or apoptotic cells in the context of KRAS mutant correction.

Response: We appreciate the reviewer's valuable comments for improving our manuscript. We additionally have examined the endogenous *KRAS* correction in HCT116 cells (human colon cancer cells bearing *KRAS* G13D mutation) and confirmed that PE4max-SpG showed up to 40.6% *KRAS* correction efficiency in HCT116. To evaluate the functional effect of *KRAS* gene correction, we measured the *KRAS* correction frequency over time and confirmed that the *KRAS* corrected cells had growth disadvantage. We have now described the results in Figure 3D-F and in the RESULTS section as below.

Figure 3D-F:

Page 6, line 25:

“In the case of HCT116, PE3b-SpG showed 32.0% KRAS correction efficiency and PEmax-SpG (hereafter named, PE4max-SpG) showed 36.1% KRAS correction efficiency (Figure 3D and 3E). We measured the KRAS correction frequency for two weeks to evaluate the functional effect of KRAS gene correction and found that the KRAS correction frequency decreased over time, reflecting that the KRAS corrected HCT116 cells have a growth disadvantage (Figure 3F).”

Minor concerns:

It is unclear what is meant by “RAS combines with guanosine triphosphatase (GTPase)”. RAS proteins are small GTPases. Please clarify what is meant here.

Response: We would like to thank the reviewer for carefully pointing out our mistake. We have now corrected the sentence as below.

Page 3, line 3:

“Ras proteins are monomeric GTPases and regulate cell differentiation, proliferation, and survival...”

The statement “KRAS has a structure lacking a pocket that can bind to allosteric inhibitors” is incorrect and is directly contradicted by literature that the authors cite in the following sentence (Ostrem et al).

Response: We would like to thank the reviewer for carefully pointing out our statment. We have now corrected incorrect expression in the sentence as below.

Page 3, line 10:

“The structure of KRAS has proven difficult to target due to its smooth surface that hinders the binding by small molecules.”

The statement “BEs can only correct transition mutations such as C:G to T:A and A:T to G:C conversion” is incorrect. Several groups have published transversion base editors. While these are not perfect tools, their omission is misleading.

Response: We appreciate the reviewer’s valuable comments for improving our manuscript. We have now revised the manuscript as below.

Page 3, line 29:

“BEs can correct transition mutations such as C:G to T:A and A:T to G:C as well as transversion mutations such as C:G to A:T or G and A:T to C:G or T:A, however, BEs can only correct cytosine or adenine within the base editing window, and often induce undesired bystander substitutions within the base editing window.”

The text suggests that CFPAC-1 cells have 4 KRAS alleles, 1 of which is KRAS G12V mutant. This would seem inconsistent with their untreated sequencing data in Figure 3F? Please reconcile this.

Response: We would like to thank the reviewer for carefully pointing out our mistake. We have now revised the sentence as below.

Page 6, line 16:

“...: two pancreatic cancer cell lines, CFPAC-1 and ASPC-1, and one colon cancer cell line, HCT116^{27,28}. We confirmed that CFPAC-1 cells have three KRAS G12V alleles and one KRAS wild-type allele, ASPC-1 cells have two KRAS G12D mutation-bearing alleles, and HCT116 cells have one KRAS G13D allele and one KRAS wild-type allele by targeted-deep sequencing.”

Figure 2 demonstrates consistent increases in prime editing efficiency through use of a PE3b strategy, including a 3.3 fold increase in the correction of the G13D mutation. The authors suggest that no mutations at the endogenous gene were observed, but do not show unintended edits for the target site. Indel frequency and unintended editing must be shown, particularly given the rate of ‘unwanted mutations’ at the edit site in Supplementary Figure 3.

Response: We appreciate the reviewer’s valuable comments for improving our manuscript. We analyzed target sites by targeted-deep sequencing and could not identify any significant “unintended edits” as well as indel mutations for the target site compared to the mock samples. To clarify the meaning, we have now modified the “unwanted mutations” to “background” in Supplementary Figure 3, which is assumed to be an error introduced during the targeted-deep sequencing process.

In Supplementary Figure 1, editing data are missing for 6 KRAS mutations.

Response: The six G13 mutations could not be targeted by the pegRNAs with *KRAS*-#1 protospacer sequences containing a 10 nt length of RTT because the distance between the nicking site and G13 mutation was more the 10 nt. We have now added “not available (n.a.)” in the Supplementary Figure 1.

Unintended editing outcomes should be shown in Supplementary Figure 2.

Response: As mentioned above, we could not identify any significant “unintended edits” in the prime editing experiment compared to the mock samples and have now added the indel frequency in the Supplementary Figure 2 as below.

Supplementary Figure 2:

It is unclear what is meant by 'unwanted mutation' in Supplementary Figure 3. Does this include indels?

Response: As mentioned above, we have now modified the "unwanted mutations" to "background", which assumed to be an error introduced during the process of targeted-deep sequencing. The indels frequency of Supplementary Figure 3 (and Figure 3C) have now summarized in Table S2.

In Supplementary Figure 3, changes in KRAS allele frequency are subtle in some cases. Appropriate statistical tests should be used to demonstrate the effect of prime editing.

Response: We have now described the prime editing frequency with biological triplicate (n=3) and summarized the prime editing frequency in Supplementary Table 2.

REVIEWERS' COMMENTS:

Reviewer #1 (Remarks to the Author):

The authors have adequately addressed my comments and answered my questions in a satisfying manner.

Reviewer #2 (Remarks to the Author):

The authors have adequately addressed the issues raised in the original review, and the manuscript is now acceptable for publication in Communications Biology.